# Collation of a century of soil invertebrate abundance data suggests long-term declines in earthworms but not tipulids

**Ailidh E. Barnes**[1]*, **Robert A. Robinson**[1], **James W. Pearce-Higgins**[1,2,3]

**1** British Trust for Ornithology, The Nunnery, Thetford, United Kingdom, **2** Conservation Science Group, Department of Zoology, University of Cambridge, Cambridge, United Kingdom, **3** School of Biological Sciences, University of East Anglia, Norwich, United Kingdom

* ailidh.barnes@bto.org

**Data Availability Statement:** The data file collated for this study is available online in the figshare repository: https://doi.org/10.6084/m9.figshare.21428121.

## Abstract

Large-scale declines in terrestrial insects have been reported over much of Europe and across the world, however, population change assessments of other key invertebrate groups, such as soil invertebrates, have been largely neglected through a lack of available monitoring data. This study collates historic data from previously published studies to assess whether it is possible to infer previously undocumented long-term changes in soil invertebrate abundance. Earthworm and tipulid data were collated from over 100 studies across the UK, spanning almost 100 years. Analyses suggested long-term declines in earthworm abundance of between 1.6 to 2.1% per annum, equivalent to a 33% to 41% decline over 25 years. These appeared greatest in broadleaved woodlands and farmland habitats, and were greater in pasture than arable farmland. Significant differences in earthworm abundance between habitats varied between models but appeared to be highest in urban greenspaces and agricultural pasture. More limited data were available on tipulid abundance, which showed no significant change over time or variation between enclosed farmland and unenclosed habitats. Declines in earthworm populations could be contributing to overall declines in ecosystem function and biodiversity as they are vital for a range of ecosystem services and are keystone prey for many vertebrate species. If robust, our results identify a previously undetected biodiversity decline that would be a significant conservation and economic issue in the UK, and if replicated elsewhere, internationally. We highlight the need for long-term and large-scale soil invertebrate monitoring, which potentially could be carried out by citizen/community scientists.

## Introduction

After Hallman et al. [1] alerted the world to large-scale declines in the abundance of aerial insects across Germany (76% over 27 years), there has been considerable interest and debate in the extent to which different invertebrate groups have declined, and the causes of any such decline. Whilst some authors highlight growing evidence for large-scale insect declines across

**Funding:** We are extremely grateful to the many BTO donors and members who have funded this project, particularly Simon Cooke and Gillian & Justin Wills and the Penchant Foundation. We are also grateful to Kenneth Trouth whose Gift in Will supported our work on this paper. The funders had no role in study design, data collection and analysis, decision to publish, or preparation of the manuscript.

**Competing interests:** The authors have declared that no competing interests exist.

the globe [2–5], others have expressed uncertainty over the extent of these declines [6, 7], in part because population trends appear to differ between taxa and studies [5]. For example, large-scale declines in pollinators [8, 9], including bees [10], macro-moths [11, 12], butterflies [13] and aerial insects [1] have been recorded, whilst other groups including emergent fresh-water insects [14] and flying aphids [12] appear to have stable or increasing populations. For many other invertebrate groups, a lack of robust monitoring data has hampered our ability to detect long-term change [7].

Whilst the current focus of the literature has been on terrestrial insect declines, other key invertebrate groups have been largely neglected, again principally due to a lack of long-term monitoring data [15]. One such neglected group are soil invertebrates, which, as important detritivores, underpin many food webs, including important prey for a wide selection of verte-brate predators [16]. They are also critically important in the delivery of a wide range of eco-system functions including nutrient cycling and soil formation [17, 18]. Despite this, soil invertebrates have been "woefully neglected in many biodiversity assessments and databases" [19]. For example, although earthworms dominate the biomass of most terrestrial ecosystems [20], their populations are not routinely monitored. This is despite them being a prime candi-date for monitoring the health of soils [21], with well-established field methods and the poten-tial for citizen science to inform large-scale earthworm monitoring [22, 23].

There is an urgent need to address this monitoring gap. Whilst it is clearly not possible to go back in time and generate monitoring data, soil invertebrates have often been the subject of ecological study generating a potentially accessible record of historical soil invertebrate abun-dance data, that may be used to infer changes through time [24]. Building on the approach of Robinson & Sutherland [25] who documented long-term declines in the availability of weed seeds on farmland from published studies, and recognising recent developments in generating large biodiversity datasets from previously published studies (e.g. van Klink [3]), we similarly attempt to identify potential long-term signals of changes in soil invertebrate abundance from published studies. The UK has a long history for ecological study, in which long-term changes in some insect populations have already been identified [26]. The fact that populations of many bird species that rely on soil invertebrates have declined, particularly on farmland and in woodland [27–31], adds weight to the suggestion that soil invertebrate populations may also be declining. Although it is likely that soil invertebrate trends are location and context dependent, and that such data are subject to a range of sampling and methodological biases, the purpose of this paper is to identify and collate such data for the UK. In doing so we focus on earthworms and tipulid larvae as the two main groups of soil macro-invertebrates in many UK soils (e.g. [32]) for which sufficient data were available for analysis, and that are also important components of the diet of many birds [16] and other taxa. Our aims are to assess, firstly, if it is possible to identify differences in soil invertebrate abundance between habitats and through time and, secondly, to test the following specific hypotheses: 1) that soil inverte-brates, particularly earthworms and tipulids, have declined in the UK and 2) that those trends vary between habitats, with the expectation of the greatest declines being in farmland, given the loss of farmland biodiversity in the UK [33].

## Methods

### Literature search

Given our focus on the UK, to maximise efficiency and avoid the need to filter tens of thou-sands of potentially useful references, we restricted our searches *a priori* to ecological journals regarded as most likely to contain relevant papers from the UK (Ibis, Bird Study, Journal of Animal Ecology, Journal of Applied Ecology, Journal of Insect Conservation, Journal of

Zoology, Biological Conservation), as well as PhD theses from UK universities, accessed via EThOS—the British Library e-theses online service (https://ethos.bl.uk/, see S1 Table for time-frame of searches). We also initially considered both the Journal of Ecology and Functional Ecology, but as initial searches found they contained very few relevant invertebrate papers, they were not considered further. We were initially uncertain how much usable data would be contained within published papers, and how well such papers might be picked-up through keyword searches. In particular, we thought that many of the most useful papers with quantitative data on the abundance of invertebrate groups would not have soil invertebrates as their main focus, but instead gather such data as covariates for studies of invertivorous species such as birds or mammals. Given these uncertainties, we started by manually scanning the contents pages of Ibis, Journal of Animal Ecology and Journal of Applied Ecology to identify potentially relevant papers rather than using keywords. This was a slow process, so having identified a range of potential papers, we then checked whether using the following combination of key-words (annelid* OR lumbricid* OR earthworm* OR insect* OR invertebrate* OR arthropod*) would be more efficient. These successfully identified 32 of 34 papers previously identified by the manual searching across 12 years of Journal of Animal Ecology and 10 years of Journal of Applied Ecology, but in less time. Therefore, this keyword approach was adopted for further journal searches. Other additional papers that were cited by these publications and seemed relevant were also considered from other journals. Potentially relevant papers and theses were identified by title and then abstract (if present), and read to check if they contained relevant and extractable/available species specific invertebrate data (i.e. they reported abundance or biomass data from locations and years). We followed the PRIMSA approach to the literature search [34], with the protocols summarised in S1 Table and the contributions of different sources to the final data shown in S1 Fig.

## Data

The following data extracted from relevant studies: reference identity, method, number of samples, area and depth of the samples, the start and end year of sampling, the season, location, description of habitat type, the units of measuring invertebrates (which varied between numbers of individuals, biomass and density in different spatial units), abundance/biomass values and what invertebrate genera data the papers contained. These were then condensed into the variables used in the analysis (Table 1). A number of different approaches to surveying soil invertebrates were used, from chemical extraction that captures individuals escaping from

**Table 1. Details of the variables collected from the relevant papers used in the analysis.**

| Variable | Explanation |
|---|---|
| *Year* (Y) | The start year of the sampling (end year was also recorded but not used in the analysis). |
| *Season* (S) | The season in which sampling occurred (Autumn, Winter, Spring, Summer and Multiple). |
| *Location* | Recorded in multiple ways in the papers but converted to GB Ordnance Survey 10 or 100 km grid reference for analysis. |
| *Sample Extent* | The area of the sample (*A*) in m$^2$ multiplied by the number of samples (*N*) to get the total sampling area/size (*Sample Extent* = A x N). |
| *Depth* (D) | The depth of the sample (cm). |
| *Method* (M) | The method of extracting soil invertebrates by chemical extraction (surface), by removing a volume of soil/turf and sorting (core) or a combination of both, split into broad (Mb) and fine (Mf) categories for separate analyses (Table 2). |
| *Habitat* (H) | The habitat in which sampling occurred split into broad (Hb) and fine (Hf) categories for separate analysis (Table 2). |

**Table 2. Factor levels in the Habitat and Method variables.** Base factor is noted first for each variable.

| Broad_Habitat | Fine_Habitat |
|---|---|
| Farmland | Arable, Pasture, Grass[1], Mixed |
| Woodland | Broadleaved, Mixed/Conifer, Scrub |
| Unenclosed | Wetland, Grassland[2], Moorland |
| Human | Industrial[3], Greenspaces |
| Broad_Method | Fine_Method |
| Core | Core |
| Both | Core and Mustard |
| Surface | Formalin, Mustard, Permanganate |

[1] Includes grass leys, field margins, meadows and set-aside

[2] Includes unenclosed and rough grassland and dunes

[3] Includes distance from factories and ex-coal mines.

the surface of the soil within a particular area [35, 36] to soil cores that extract individuals from a particular volume (area x depth) of soil [27, 37] and summarised into different categories as they will have different efficacies that need to be accounted for in the analysis. Sufficient data for subsequent analysis were available for Lumbricidae (earthworm species) and Tipulidae larvae (leatherjackets) only. Although most studies provided overall estimates of abundance and/or biomass, some separated these into adult and immature numbers/biomass, or presented only one metric. If provided separately, both (adult and immature numbers) were summed to gain total earthworm numbers. Where total abundance data were missing but either adult numbers or wet biomass (for adults and/or immatures) data were provided alone, total abundance was estimated from the provided data using regressions of total abundance against adult abundance (Eq 1), and total abundance against biomass (Eq 2) based on data from studies where both were presented (S2 and S3 Figs in S1 File). Eq 2 was calculated only using adult wet weights/biomass (in grams) but was used to calibrate both adult and immature wet weights/biomass into total earthworm abundance. A small number of studies reported earthworm dry biomass, but correlations were not good enough to calibrate these against earthworm numbers (Total Number of Earthworms = 85 + 1.9 x Dry Biomass, R = 0.31, n = 37, P = 0.13), and so these data were not included in our analysis. Given the variable nature of the data reported, with some studies presenting only densities and others total numbers, all abundances were standardised to density per $m^2$ for analysis. As earthworm abundance varies with sample/soil depth, this was included in the models.

$$Total\ Abundance\ of\ Earthworms\ (m^{-2}) = 46 + 2\ x\ Number\ of\ Adult\ Earthworms\ (m^{-2})\quad(1)$$

($R^2 = 0.78$, n = 295, P <0.001)

$$Total\ Abundance\ of\ Earthworms\ (m^{-2}) = 85 + 2\ x\ Wet\ Biomass\ (gm^{-2})\quad(2)$$

($R^2 = 0.56$, n = 370, P <0.001)

## Analysis

The abundances (per $m^2$) of earthworms ($A_E$) and tipulids ($A_T$) rounded to the nearest integer, were modelled as functions of five explanatory variables; *Year* (Y), *Habitat* (H), *Method* (M), *Season* (S) and *Depth* (D; see Table 1). The latter four variables were included to control for

potentially important differences between studies and used in various combinations to answer the following questions for each taxon:

i.  Has abundance changed through time? Assessed using *Year* as a continuous variable (Models 1 and 2).

ii.  Does abundance vary with habitat? Comparing variation between *Broad_Habitat* categories (Model 1) and *Fine_Habitat* categories (Model 2).

$$\text{Model 1}: A_{E/T} = Y + Hb + Mb + S + D$$

$$\text{Model 2}: A_E = Y + Hf + Mb + S + D$$

iii.  Do abundance trends vary with habitat? Model 3 tests the interaction between *Broad_Habitat* and *Year* as a continuous variable and Model 4 tests the interaction between *Fine_Habitat* and *Year*.

$$\text{Model 3}: A_{E/T} = Y + Hb + Y*Hb + Mb + S + D$$

$$\text{Model 4}: A_E = Y + Hf + Y*Hf + Mb + S + D$$

iv.  Does the method of extraction affect abundance? Models 5 to 8 are equivalents to models 1 to 4 but using *Fine_Method* categories. Note that it was not possible to categorise some studies by *Fine_Method*, so models 5 to 8 are associated with a smaller sample size than models 1 to 4. For this reason, we focus on the analysis of *Broad_Method* categories in the results.

$$\text{Model 5}: A_E = Y + Hb + Mf + S + D$$

$$\text{Model 6}: A_E = Y + Hf + Mf + S + D$$

$$\text{Model 7}: A_E = Y + Hb + Y*Hb + Mf + S + D$$

$$\text{Model 8}: A_E = Y + Hf + Y*Hf + Mf + S + D$$

The intercept for *Year* was set at 1960, so that the intercept of the fitted relationship represented the predicted abundance in 1960, which was the beginning of the period from which most data were collected. *Method* and *Habitat* categories with fewer than six studies were excluded from the respective fine-scale analyses. The glmmTMB function and package in R (version 4.0.3; [38, 39]) was used to generate a generalised linear mixed model (glmm) using a negative binomial error function, to account for overdispersion (theta was 98 for the earthworm and 111 for tipulid unweighted Poisson models). To account for the non-independence of results from the same study and similar locations, both *Study_ID* and *Location* were included as random factors. *Location* was specified by either the 10 or 100 km grid reference within which a study was located—the latter for more dispersed studies, or studies where a clear location was not reported (e.g. a county).

We were keen to account for the fact that some studies were associated with more certain estimates of abundance than others, but too few studies presented variance in estimates of soil invertebrate abundance in a consistent manner for us to do this easily. Instead of measuring variance directly, we might expect that studies with a large *Sample Extent* are likely to be less stochastic and therefore can potentially be treated with more confidence, we therefore used the natural log plus one of *Sample Extent* as a weighting variable in the analysis. However, it is equally possible that such studies could represent a biased subset of studies, particularly given a tendency for older studies to be more extensive (Appendix 1 in S1 File). To account for this, we repeated each model without weighting. We followed the recommendations of Kotze et al. [40] by including information on seasonal variation in detectability, and by specifying the response variable as following a negative binomial distribution, as noted earlier. As we were modelling reported standardised densities rather than count data, due to the variable nature of the data (see earlier), we were unable to follow Kotze et al. [40] and use an offset with count (to give density), although the unweighted model results, based on density values, is equivalent to this approach. To capture any uncertainty we present both the weighted and unweighted results, but for simplicity, most graphs present the non-weighted estimates.

## Results

### Characteristics of published studies

Following our search protocols (S1 Fig and S1 Table), we identified 104 studies containing relevant data, comprising 70 peer-reviewed papers, 3 books and book chapters and 31 theses. A total of 90 studies contained earthworm data from 113 locations and 29 studies contained tipulid data from 42 locations (Fig 1). In total these comprised a maximum of 2,210 individual earthworm data rows and 746 tipulid data rows for inclusion in Model 1 (the simplest). Studies spanned almost 100 years from 1920 to 2018; the latest date for tipulids was 2011. The median time span per earthworm study was 1 year and 2 years for tipulids (and the longest study 13 and 26 years, respectively), most were single year studies. The majority of data were from farm-land for both taxa (Fig 2), which comprised the greatest spread of data through time, compared to the other *Broad_Habitats*, although, with the exception of tipulid data from unenclosed habitats, data spanned at least 5 decades. Given the more limited data, tipulids were only modelled in relation to broad *Habitat* and *Method* categories (Models 1 and 3). Core sampling was the most common method of data collection for both groups (Fig 2).

### Modelling

**Earthworms.** Changes in earthworm abundance from Model 1 were suggestive of an overall decline through time (Fig 3). The coefficient with *Year* was of a similar magnitude and significance in both the unweighted (-0.017 ± 0.006, P = 0.01) and weighted models (-0.017 ± 0.006, P = 0.01), suggesting this decline is robust to variation in sample-size between studies. Focussing on the unweighted model, there were significantly fewer earthworms in samples in the summer than the autumn, and using chemical (Surface) and both surface and core samples combined than by using cores, whilst there was a positive relationship between *Depth* and earthworm abundance (Table 3); deeper cores contained more earthworms.

After accounting for these methodological differences, there was no significant difference in earthworm density between the *Broad_Habitat* categories in the unweighted model, although densities were slightly lower in woodland (Fig 4a). This differs slightly from the weighted model, where densities were significantly reduced in unenclosed and woodland habitats compared to the base/intercept category of farmland (Table 3 and Fig 4b). The unweighted Model 2 showed significant differences between some of the *Fine_Habitat* categories (Table 4).

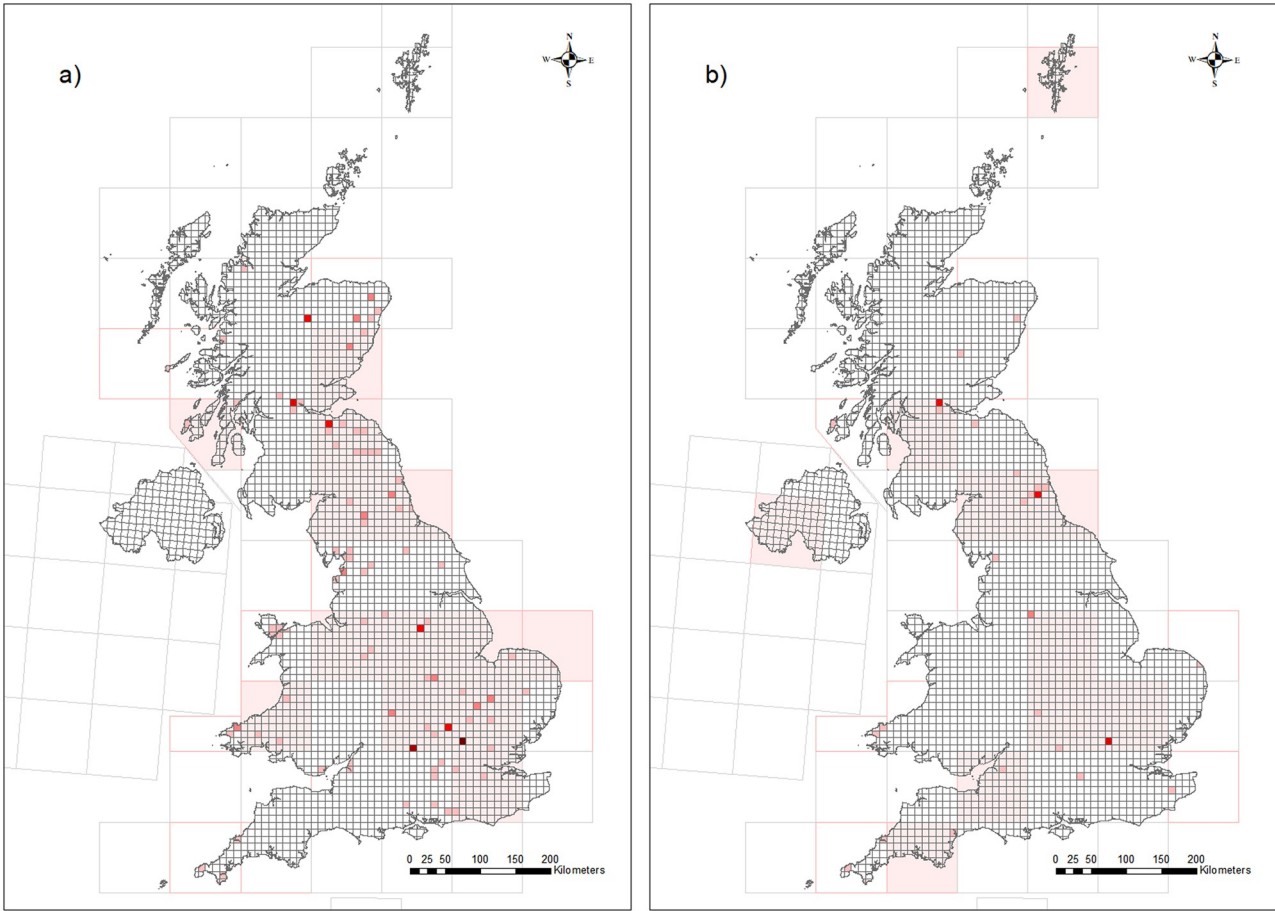

**Fig 1. The geographic spread of a) earthworm data and b) tipulid data extracted from the literature over the UK.** The red 10 km squares represent locations sampled for each study and darker squares represent data from multiple studies. The light red 100 km squares are either inaccurate study locations or studies that cover a wider area than the 10 km. © 2018–2022 GADM—license. a) b).

Pasture had a significantly higher earthworm abundance than arable (Table 4), but the highest earthworm abundances were in unenclosed greenspace and the lowest abundances were in the industrial habitats (Table 4 and Fig 5). When sample extent was taken into account in the weighted model, the lowest earthworm abundances were associated with moorland, but abundances in industrial habitats did not differ significantly from arable (Table 4).

Including the interaction between *Year* and *Broad_Habitat* (Model 3; Table 5) showed that the strongest evidence for negative trends were restricted to the base farmland (-0.016 ± 0.006, P = 0.01) and woodland (-0.043 ± 0.010, P <0.001) habitats (Fig 6). Trends in farmland were significantly more negative than the non-significant positive trends in unenclosed habitats, and those in woodland were significantly more negative than either farmland or unenclosed habitats (Table 5). The weighted model showed similarly declining trends in farmland and woodland habitats (Table 5).

Investigating the finer habitat categories (*Fine_Habitat*) in Model 4 revealed that the declines in woodland were from broadleaved woodland (-0.057 ± 0.013, P = <0.001), with no overall trend apparent from mixed/conifer woodland (0.012 ± 0.016, P = 0.27; Table 6 and Fig7). Declines on farmland were particularly apparent from pasture (-0.014 ± 0.006, P = 0.04) and grass categories (-0.015 ± 0.005, P = 0.06), compared to marginal declines on arable

(a)

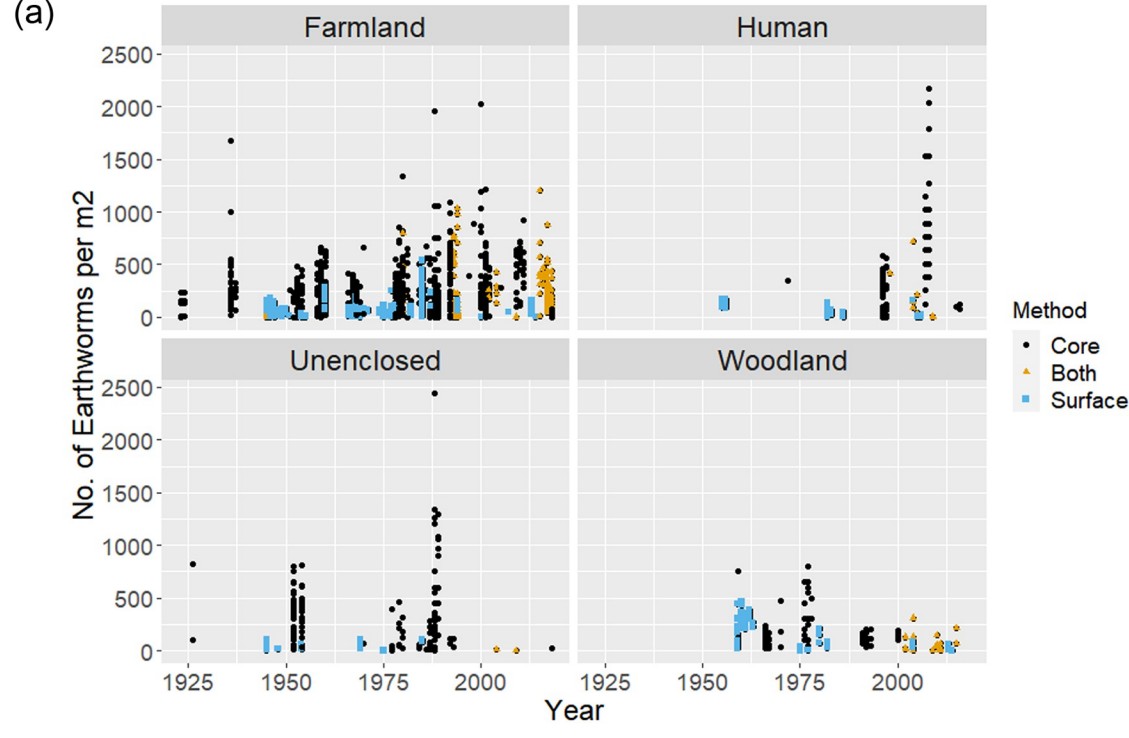

(b)

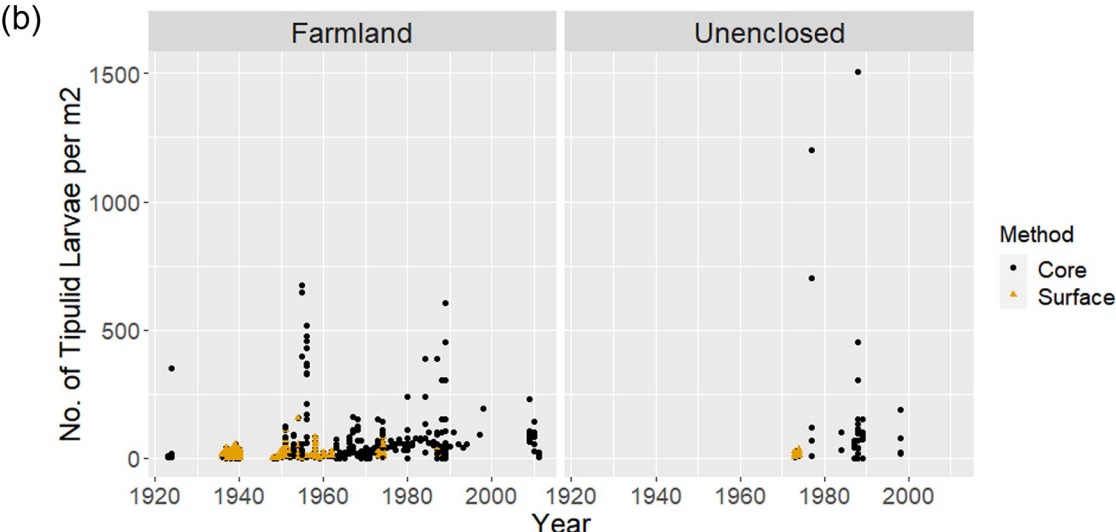

**Fig 2. The temporal spread of a) earthworm data and b) tipulid data.** Plotted per m² across the Broad_Habitat categories and coloured by the Broad_Method categories. Sufficient tipulid data existed for analysis of unenclosed and farmland habitats only.

habitats (base factor; 0.006 ± 0.007, P = 0.48) and marginal increases on mixed farmland (0.041 ± 0.035, P = 0.25). Both human *Fine_Habitat* categories showed evidence of decline (greenspace -0.048 ± 0.020, P = 0.01; industrial -0.063 ± 0.031, P = 0.04; Fig 7), whilst there was no evidence of significant change across unenclosed habitats (Fig 7).

Providing a finer separation of the different methods of soil invertebrate extraction (*Fine_Method*) in models 5–8 reiterates that a greater abundance of earthworms were collected using core sampling than surface sampling (S3 Fig in S1 File), and broadly supported

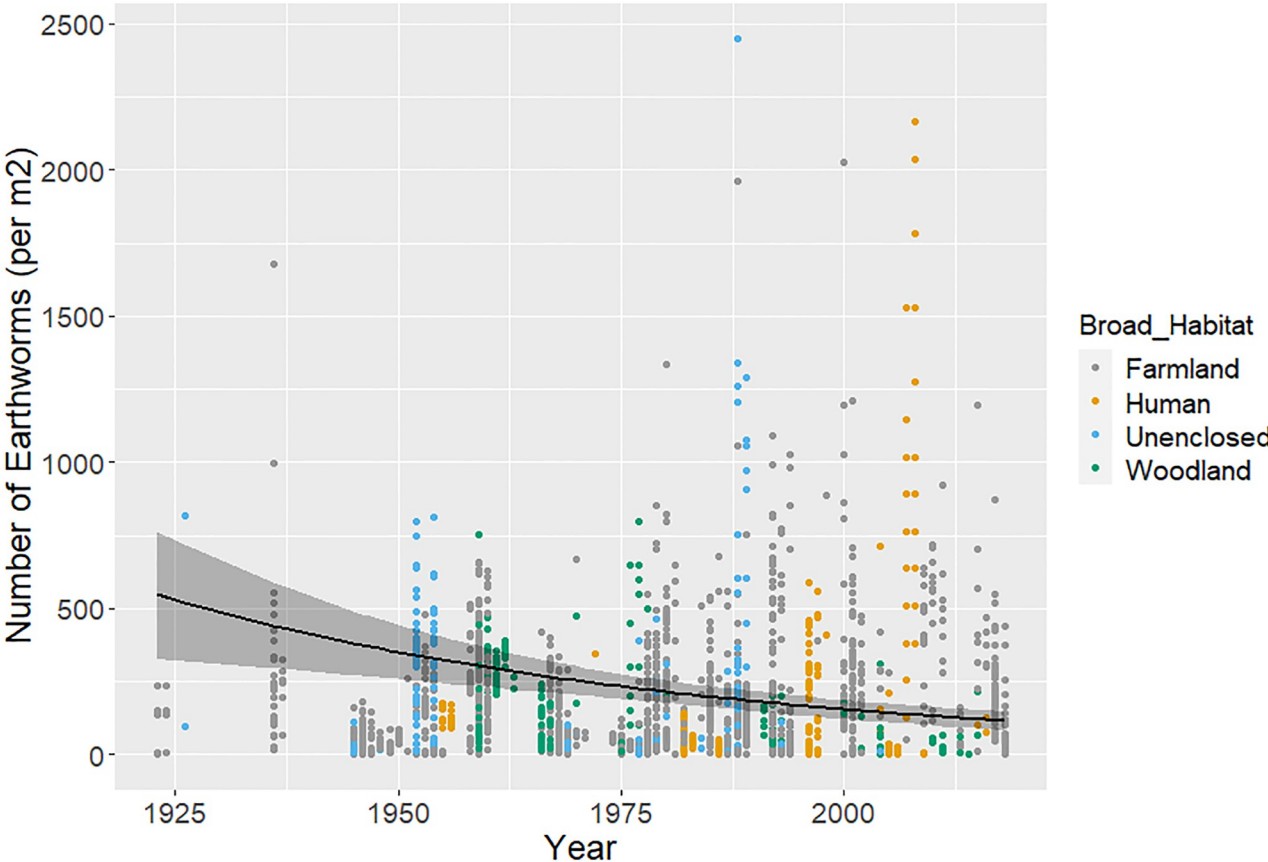

**Fig 3. Overall earthworm trend.** Overall year trend from unweighted Model 1 with standard error, coloured by Broad_Habitat.

**Table 3. Model 1 output of earthworm abundance.** Unweighted and weighted (by the natural log of Sample Extent + 1) Model 1 estimates without the interaction, compared to the base of Farmland, Autumn and Core (significant estimates are in bold, the stars represent: * <0.01, ** <0.001, *** <0.0001, SE = Standard Error).

| Variable Category | Variable | Unweighted | | | Weighted | | |
|---|---|---|---|---|---|---|---|
| | | Estimate | SE | P | Estimate | SE | P |
| | (Intercept) | **5.666** | **0.227** | **<0.001 (***)** | **5.868** | **0.233** | **<0.001 (***)** |
| | year | **-0.016** | **0.006** | **0.006 (**)** | **-0.017** | **0.006** | **0.009 (**)** |
| *Broad_Habitat* | Human | -0.007 | 0.226 | 0.976 | -0.117 | 0.196 | 0.551 |
| | Unenclosed | -0.009 | 0.126 | 0.944 | **-0.870** | **0.169** | **<0.001 (***)** |
| | Woodland | -0.245 | 0.145 | 0.091 | **-0.400** | **0.135** | **0.003 (**)** |
| *Season* | Multiple | **0.330** | **0.145** | **0.023 (*)** | **0.458** | **0.115** | **<0.001 (***)** |
| | Spring | 0.033 | 0.061 | 0.592 | -0.012 | 0.059 | 0.836 |
| | Summer | **-0.373** | **0.069** | **<0.001 (***)** | **-0.248** | **0.068** | **<0.001 (***)** |
| | Winter | -0.031 | 0.073 | 0.672 | -0.052 | 0.078 | 0.511 |
| *Broad_Method* | Both | **-0.836** | **0.231** | **<0.001 (***)** | **-0.784** | **0.207** | **<0.001 (***)** |
| | Surface | **-1.333** | **0.157** | **<0.001 (***)** | **-1.334** | **0.156** | **<0.001 (***)** |
| | Depth | **0.011** | **0.005** | **0.041 (*)** | 0.000 | 0.005 | 0.922 |

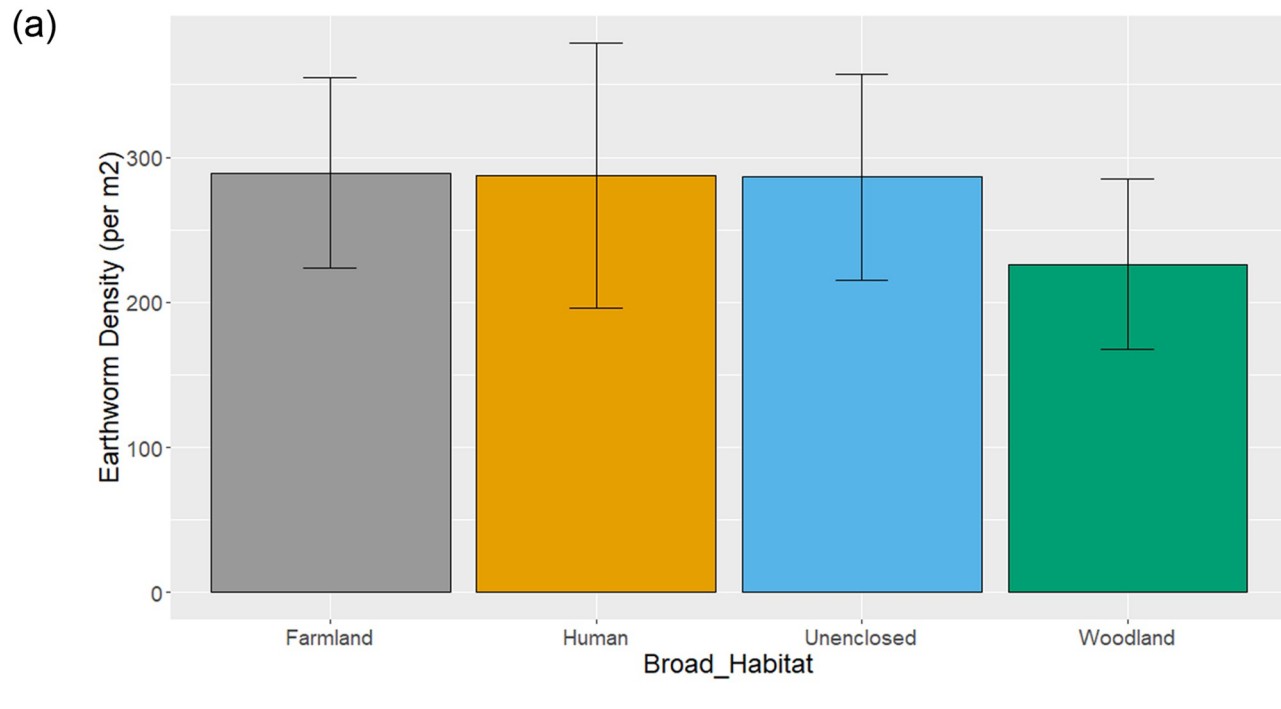

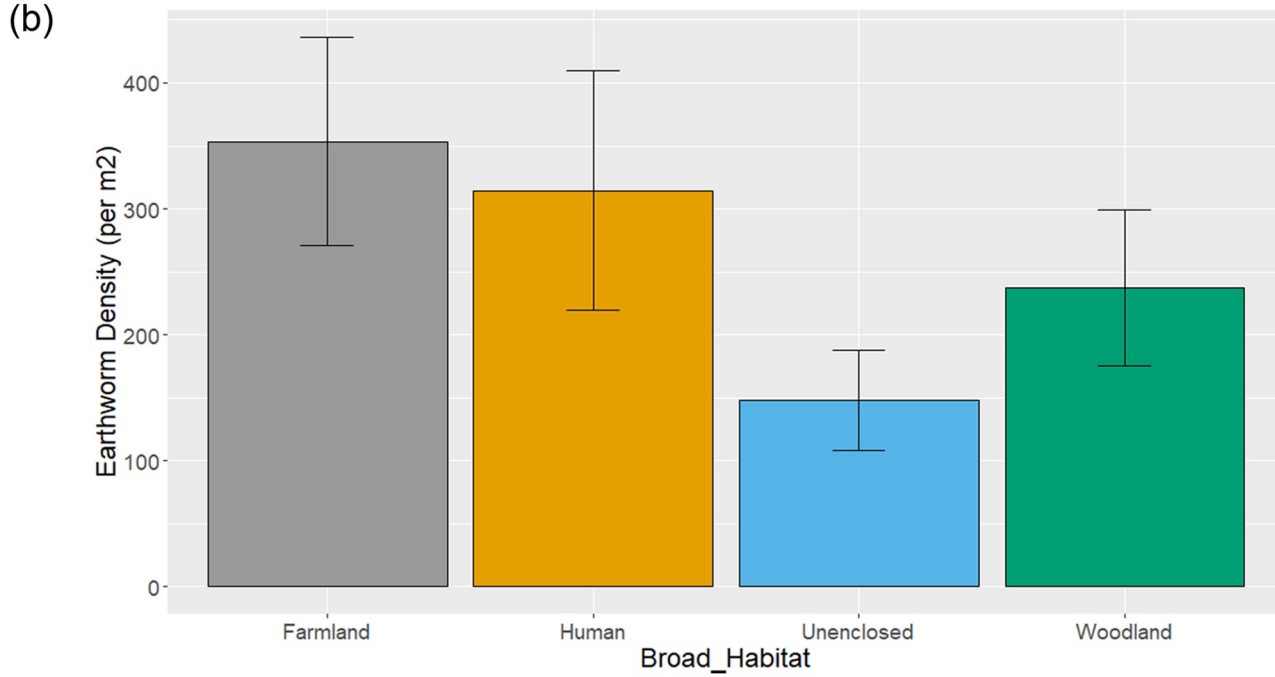

**Fig 4. Broad_Habitat specific earthworm density.** Average earthworm density per m² in each Broad_Habitat category from a) the unweighted and b) weighted (by natural log of Sample Extent + 1) using Model 1. See Table 3 for estimates.

the results of the previous analyses. The one exception was with respect to abundance in industrial habitats probably due to the reduced sample size of this analysis (S4-S6 Figs and S3-S6 Tables in S1 File).

**Tipulids.** There was less consistency between the weighted and unweighted tipulid models than for earthworms, suggesting that the reduced sample size for this group made the

**Table 4. Model 2 output of earthworm abundance.** Estimates from Fine_Habitat Model 2, unweighted and weighted (by natural log of Sample Extent + 1), compared to the base of Farmland_Arable, Autumn and Core. The level of significance is represented by: * p < 0.05; ** p < 0.01; *** p < 0.001, SE = Standard Error.

| Variable Category | Variable | Unweighted | | | Weighted | | |
|---|---|---|---|---|---|---|---|
| | | Estimate | SE | P | Estimate | SE | P |
| | (Intercept) | **5.406** | **0.243** | **<0.001 (***)** | **5.517** | **0.255** | **<0.001 (***)** |
| | year | **-0.016** | **0.006** | **0.007 (**)** | **-0.016** | **0.006** | **0.014 (*)** |
| *Fine_Habitat* | Woodland_Broadleaved | 0.299 | 0.193 | 0.121 | 0.375 | 0.195 | 0.054 |
| | Farmland_Grass | 0.016 | 0.130 | 0.904 | 0.068 | 0.146 | 0.642 |
| | Unenclosed_Grassland | 0.219 | 0.170 | 0.197 | -0.180 | 0.213 | 0.399 |
| | Human_Greenspace | **0.781** | **0.288** | **0.007 (**)** | **0.530** | **0.266** | **0.046 (*)** |
| | Human_Industrial | **-0.854** | **0.317** | **0.007 (**)** | -0.497 | 0.319 | 0.119 |
| | Farmland_Mixed | 0.342 | 0.440 | 0.436 | 0.311 | 0.330 | 0.347 |
| | Woodland_Mixed/Conifer | **-0.449** | **0.203** | **0.027 (*)** | **-0.657** | **0.186** | **0.0004 (***)** |
| | Unenclosed_Moorland | **0.561** | **0.191** | **0.003 (**)** | **-1.459** | **0.303** | **<0.001 (***)** |
| | Farmland_Pasture | **0.377** | **0.115** | **0.001 (**)** | **0.488** | **0.128** | **<0.001 (***)** |
| | Woodland_Scrub | 0.164 | 0.600 | 0.785 | 0.027 | 1.150 | 0.981 |
| | Unenclosed_Wetland | -0.078 | 0.281 | 0.781 | -0.720 | 0.612 | 0.239 |
| *Season* | Multiple | **0.350** | **0.144** | **0.015 (*)** | **0.471** | **0.114** | **<0.001 (***)** |
| | Spring | 0.039 | 0.060 | 0.520 | -0.016 | 0.058 | 0.783 |
| | Summer | **-0.359** | **0.069** | **<0.001 (***)** | **-0.248** | **0.067** | **<0.001 (***)** |
| | Winter | -0.020 | 0.072 | 0.786 | -0.049 | 0.077 | 0.528 |
| *Broad_ Method* | Both | **-0.825** | **0.230** | **<0.001 (***)** | **-0.746** | **0.205** | **<0.001 (***)** |
| | Surface | **-1.348** | **0.155** | **<0.001 (***)** | **-1.335** | **0.154** | **<0.001 (***)** |
| | Depth | **0.011** | **0.005** | **0.032 (*)** | 0.001 | 0.005 | 0.901 |

results more sensitive to the assumptions made. The unweighted model suggests abundances have been broadly similar through time, whilst the weighted model suggests that they may actually have increased (Table 7). Critically, neither model suggested that tipulid abundances have declined. The unweighted model suggested higher abundances were found using core sampling than surface sampling (-2.139 ± 0.618, P = 0.001) and with greater abundances in all seasons compared to the autumn, but particularly in the spring (1.999 ± 0.328, P <0.0001).

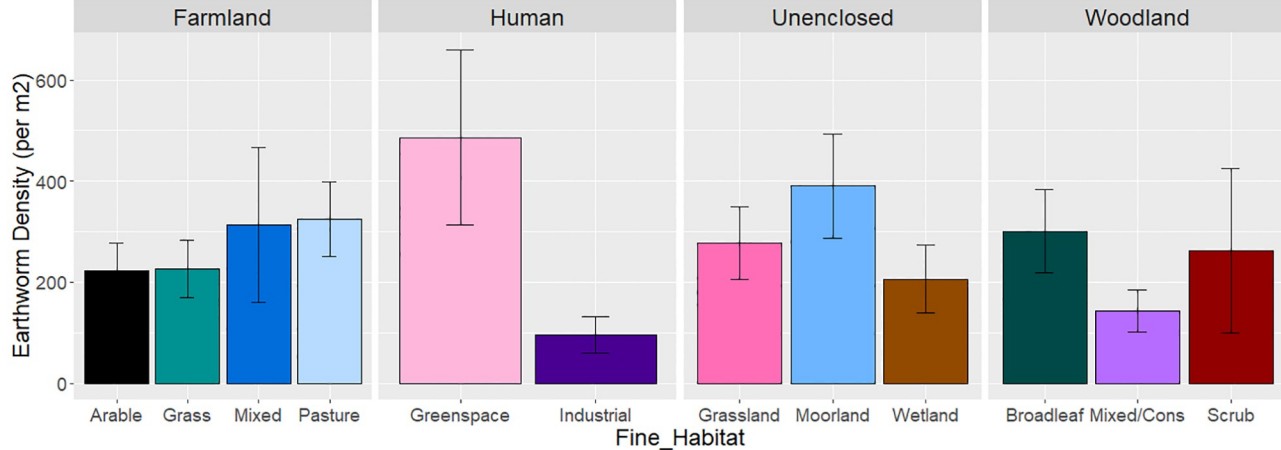

**Fig 5. Fine_Habitat specific earthworm density.** Average earthworm density per m² for each Fine_Habitat category from Model 2 (unweighted). See Table 4 for estimates.

**Table 5. Model 3 output of earthworm abundance trends.** Unweighted and weighted (by the natural log of Sample Extent +1) Model 3 estimates with the interaction of Year with Broad_Habitat providing habitat specific trends compared to the base, farmland, trend shown in the year estimate (significant estimates are in bold, the stars represent: * <0.01, ** <0.001, *** <0.0001, SE = Standard Error).

| Variable Category | Variable | Unweighted | | | Weighted | | |
|---|---|---|---|---|---|---|---|
| | | Estimate | SE | P | Estimate | SE | P |
| | (Intercept) | **5.686** | **0.234** | **<0.001 (***)** | **5.818** | **0.232** | **<0.001 (***)** |
| | year | **-0.016** | **0.006** | **0.011 (*)** | **-0.015** | **0.006** | **0.021 (*)** |
| *Broad_Habitat* | Human | 0.051 | 0.566 | 0.928 | 0.004 | 0.507 | 0.994 |
| | Unenclosed | -0.526 | 0.277 | 0.057 | **-0.951** | **0.183** | **<0.001 (***)** |
| | Woodland | 0.615 | 0.361 | 0.088 | 0.275 | 0.283 | 0.332 |
| *Season* | Multiple | **0.339** | **0.145** | **0.020 (*)** | **0.456** | **0.115** | **<0.001 (***)** |
| | Spring | 0.039 | 0.061 | 0.524 | -0.009 | 0.059 | 0.884 |
| | Summer | **-0.370** | **0.069** | **<0.001 (***)** | **-0.246** | **0.068** | **<0.001 (***)** |
| | Winter | -0.028 | 0.073 | 0.699 | -0.051 | 0.078 | 0.516 |
| *Broad_Method* | Both | **-0.870** | **0.229** | **<0.001 (***)** | **-0.799** | **0.205** | **<0.001 (***)** |
| | Surface | **-1.366** | **0.156** | **<0.001 (***)** | **-1.337** | **0.155** | **<0.001 (***)** |
| | Depth | **0.011** | **0.005** | **0.046 (*)** | 0.000 | 0.005 | 0.934 |
| *Broad_Habitat* specific Trends | Human | -0.003 | 0.016 | 0.833 | -0.005 | 0.014 | 0.700 |
| | Unenclosed | **0.021** | **0.010** | **0.037 (*)** | 0.009 | 0.008 | 0.253 |
| | Woodland | **-0.027** | **0.010** | **0.009 (**)** | **-0.022** | **0.008** | **0.006 (**)** |

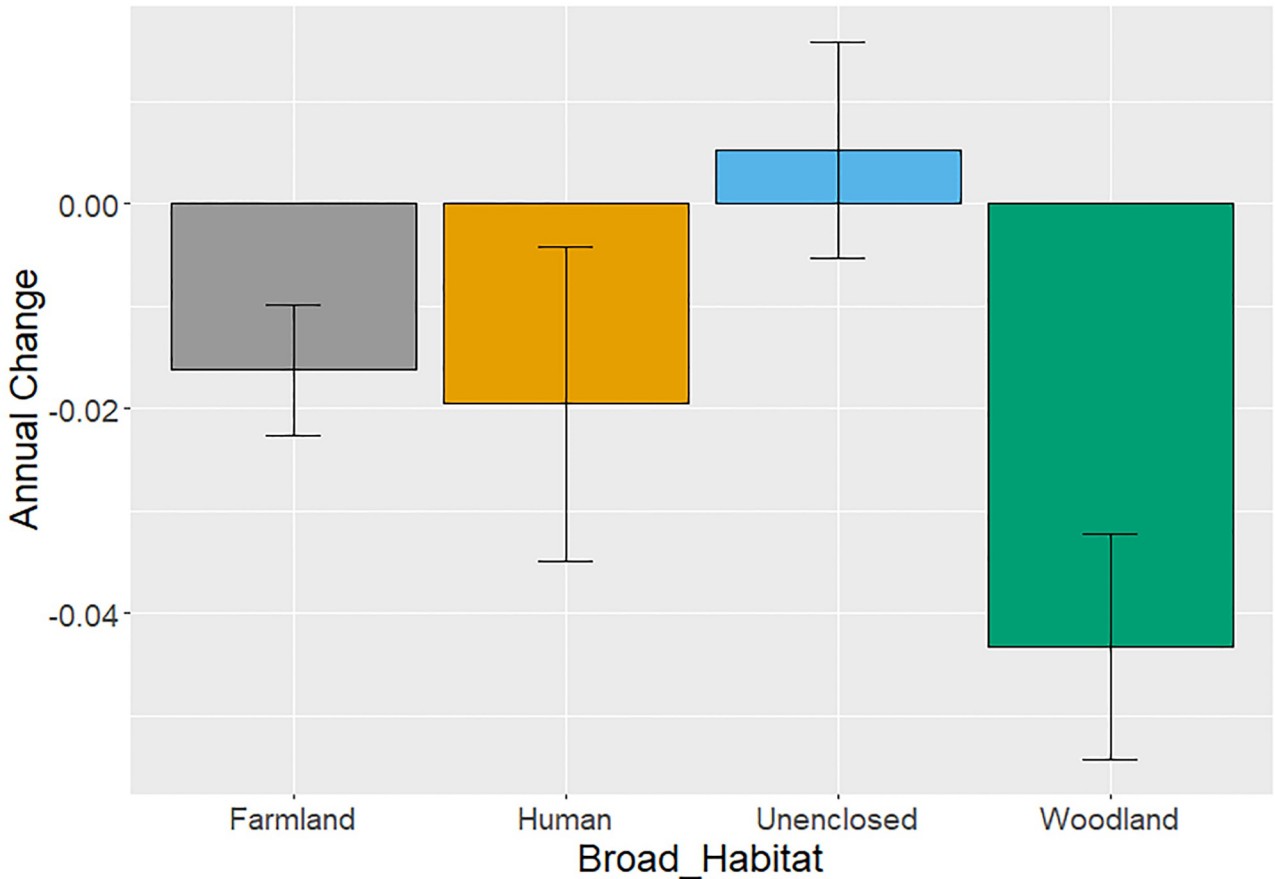

**Fig 6. Broad_Habitat specific trends in earthworm abundance.** Habitat specific earthworm abundance trends using unweighted Model 3 including the interaction with Year and Broad_Habitat. See Table 5 for estimates (the stars represent significance from zero: * <0.01, ** <0.001, *** <0.0001).

**Table 6. Model 4 output of earthworm abundance trends.** The output from Model 4 using the Fine_Habitat categories with the interaction with year, unweighted and weighted (by the natural log of Sample Extent +1), compared to the base of Farmland_Arable, Autumn and Core. The level of significance is represented by:. p < 0.1; * p < 0.05; ** p < 0.01; *** p < 0.001, SE = Standard Error.

| | Variable | Unweighted | | | Weighted | | |
|---|---|---|---|---|---|---|---|
| | | Estimate | SE | P | Estimate | SE | P |
| | (Intercept) | **5.082** | **0.289** | **<0.001 (***)** | **5.245** | **0.280** | **<0.001 (***)** |
| | year | -0.006 | 0.007 | 0.416 | -0.005 | 0.007 | 0.476 |
| *Fine_ Habitat* | Woodland_Broadleaved | **1.792** | **0.436** | **<0.001 (***)** | **1.519** | **0.384** | **<0.001 (***)** |
| | Farmland_Grass | 0.307 | 0.226 | 0.173 | 0.300 | 0.178 | 0.091 (.) |
| | Unenclosed_Grassland | 0.081 | 0.373 | 0.828 | -0.092 | 0.283 | 0.745 |
| | Human_Greenspace | **2.254** | **0.850** | **0.008 (**)** | **2.151** | **0.865** | **0.013 (*)** |
| | Human_Industrial | 1.041 | 1.015 | 0.305 | 1.815 | 0.947 | 0.055 (.) |
| | Farmland_Mixed | -1.455 | 1.392 | 0.296 | **-2.801** | **0.955** | **0.003 (**)** |
| | Woodland_Mixed/Conifer | -1.168 | 0.621 | 0.060 (.) | **-1.212** | **0.465** | **0.009 (**)** |
| | Unenclosed_Moorland | 0.477 | 0.452 | 0.291 | **-1.295** | **0.351** | **<0.001 (***)** |
| | Farmland_Pasture | **0.697** | **0.241** | **0.004 (**)** | **0.661** | **0.213** | **0.002 (**)** |
| | Woodland_Scrub | 1.064 | 2.531 | 0.674 | 2.942 | 3.329 | 0.377 |
| | Unenclosed_Wetland | -0.122 | 0.483 | 0.801 | -0.511 | 0.607 | 0.400 |
| *Season* | Multiple | **0.370** | **0.143** | **0.010 (*)** | **0.477** | **0.113** | **<0.001 (***)** |
| | Spring | 0.038 | 0.060 | 0.530 | -0.023 | 0.058 | 0.697 |
| | Summer | **-0.364** | **0.069** | **0.000** | **-0.257** | **0.067** | **0.001 (***)** |
| | Winter | -0.021 | 0.072 | 0.768 | -0.057 | 0.077 | 0.463 |
| *Broad_ Method* | Both | **-0.893** | **0.227** | **<0.001 (***)** | **-0.787** | **0.202** | **<0.001 (***)** |
| | Surface | **-1.413** | **0.154** | **<0.001 (***)** | **-1.359** | **0.153** | **<0.001 (***)** |
| | Depth | **0.011** | **0.005** | **0.045 (*)** | 0.000 | 0.004 | 0.952 |
| *Fine_ Habitat* specific Trends | Woodland_Broadleaved | **-0.051** | **0.013** | **0.001 (***)** | **-0.050** | **0.013** | **<0.001 (***)** |
| | Farmland_Grass | -0.009 | 0.005 | 0.116 | **-0.011** | **0.005** | **0.017 (*)** |
| | Unenclosed_Grassland | 0.010 | 0.013 | 0.442 | 0.004 | 0.011 | 0.741 |
| | Human_Greenspace | **-0.042** | **0.020** | **0.033 (*)** | **-0.049** | **0.020** | **0.013 (*)** |
| | Human_Industrial | -0.057 | 0.031 | 0.068 (.) | **-0.080** | **0.032** | **0.013 (*)** |
| | Farmland_Mixed | 0.047 | 0.035 | 0.182 | **0.069** | **0.022** | **0.001 (**)** |
| | Woodland_Mixed/Conifer | 0.018 | 0.016 | 0.273 | 0.010 | 0.011 | 0.373 |
| | Unenclosed_Moorland | 0.007 | 0.016 | 0.656 | -0.025 | 0.014 | 0.065 (.) |
| | Farmland_Pasture | -0.008 | 0.006 | 0.173 | -0.005 | 0.006 | 0.372 |
| | Woodland_Scrub | -0.033 | 0.086 | 0.699 | -0.127 | 0.129 | 0.324 |
| | Unenclosed_Wetland | 0.011 | 0.018 | 0.530 | 0.016 | 0.024 | 0.505 |

The weighted model highlighted similar patterns, but with no significant differences with *Broad_Method* categories and the lowest tipulid abundance in the Summer, not the Autumn. Neither model identified significant variation in abundance between unenclosed and farmland habitats, or in trend between habitats (Table 7 and Fig 8).

## Discussion

Despite their economic, ecological and biodiversity importance, soil invertebrates are generally poorly monitored, including in the UK. To start to address this gap, in an attempt to reconstruct potential past trends, we identified over 100 ecological studies from literature in the UK that contained quantitative data on earthworm and tipulid abundances over a span of almost 100 years, although the majority of the data were collected post 1960. While the data

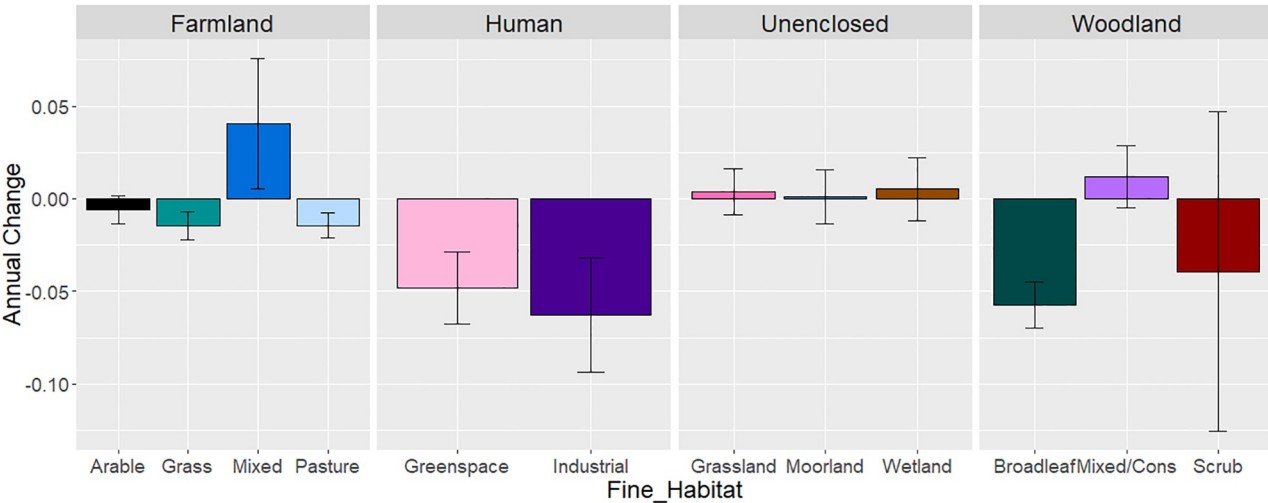

**Fig 7. Fine_Habitat specific earthworm abundance trends.** Earthworm abundance trends from the interaction of Year with each Fine_Habitat category from Model 4 (unweighted). The level of significance from zero is indicated by:. p < 0.1; * p < 0.05; ** p < 0.01; *** p < 0.001. See Table 6 for estimates.

**Table 7. Model 1 and 3 output of tipulid abundance and trends.** Estimates from the tipulid models weighted (by the natural log of Sample Extent +1) and unweighted. Top is Model 1 and bottom is Model 3 with an interaction between Year and Broad_Habitat, compared to the base of Farmland, Core and Autumn. Significant estimates are in bold, the stars represent: * <0.01, ** <0.001, *** <0.0001, SE = Standard Error.

| Variable Category | Variable | Unweighted | | | Weighted | | |
|---|---|---|---|---|---|---|---|
| | | **Estimate** | **SE** | **P** | **Estimate** | **SE** | **P** |
| | (Intercept) | **3.369** | **0.665** | **<0.001 (***)** | **3.356** | **0.866** | **<0.001 (***)** |
| | year | 0.000 | 0.012 | 0.988 | **0.023** | **0.011** | **0.041 (*)** |
| *Broad_Habitat* | Unenclosed | 0.214 | 0.282 | 0.446 | -0.001 | 0.436 | 0.997 |
| *Broad_Method* | Surface | **-2.139** | **0.618** | **0.001 (***)** | -0.889 | 0.881 | 0.313 |
| | Depth | -0.052 | 0.055 | 0.343 | 0.025 | 0.065 | 0.699 |
| *Season* | Multiple | **1.662** | **0.500** | **0.001 (***)** | 0.930 | 0.507 | 0.067 |
| | Spring | **1.999** | **0.328** | **<0.001 (***)** | 0.863 | 0.471 | 0.067 |
| | Summer | **0.731** | **0.304** | **0.016 (*)** | **-1.239** | **0.461** | **0.007 (**)** |
| | Winter | **0.797** | **0.336** | **0.018 (*)** | 0.560 | 0.516 | 0.278 |
| **Interaction Model** | | **Estimate** | **SE** | **P** | **Estimate** | **SE** | **P** |
| | (Intercept) | **3.380** | **0.663** | **<0.001 (***)** | **3.379** | **0.874** | **<0.001 (***)** |
| | year | 0.0004 | 0.012 | 0.973 | **0.024** | **0.012** | **0.041 (*)** |
| *Broad_Habitat* | Unenclosed | 0.542 | 1.282 | 0.672 | 0.280 | 1.184 | 0.813 |
| *Broad_Method* | Surface | **-2.143** | **0.616** | **<0.001 (***)** | -0.910 | 0.889 | 0.306 |
| | Depth | -0.053 | 0.055 | 0.335 | 0.024 | 0.066 | 0.720 |
| *Season* | Multiple | **1.642** | **0.505** | **0.001 (**)** | 0.926 | 0.507 | 0.068 |
| | Spring | **1.994** | **0.330** | **<0.001 (***)** | 0.861 | 0.472 | 0.068 |
| | Summer | **0.729** | **0.305** | **0.017 (*)** | **-1.242** | **0.462** | **0.007 (**)** |
| | Winter | **0.793** | **0.337** | **0.019 (*)** | 0.558 | 0.516 | 0.280 |
| *Broad_Habitat* Interaction | Unenclosed | -0.013 | 0.048 | 0.793 | -0.013 | 0.052 | 0.798 |

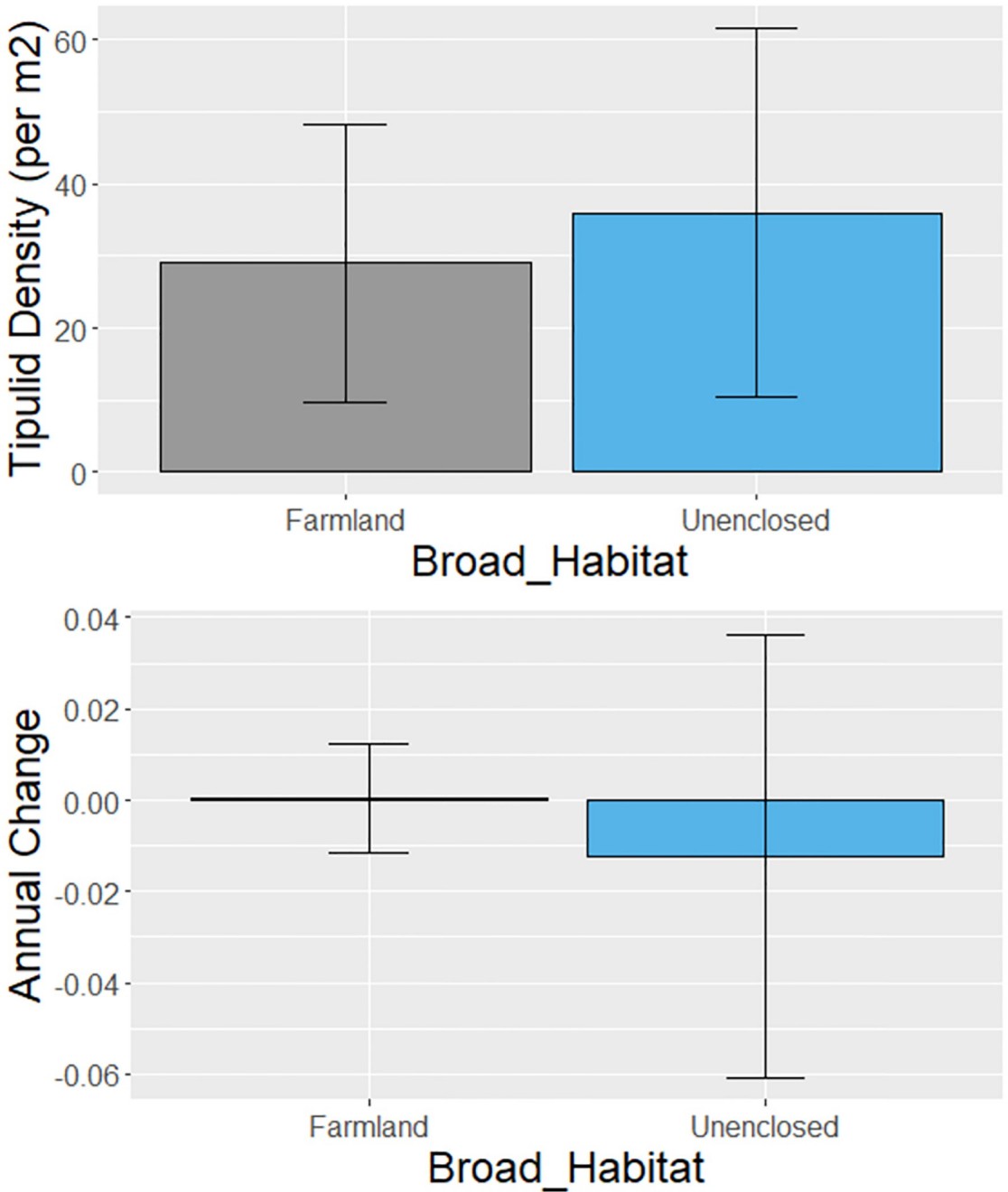

**Fig 8. Broad_Habitat specific tipulid density and trend.** Average tipulid density per m$^2$ from unweighted Model 1 (above) and trend estimates (below, not significant) from interaction of unweighted Model 3, for each Broad_Habitat. See Table 7 for estimates.

are unstructured, analyses suggest a signal consistent with earthworm abundances having declined in the UK since 1960. In contrast, we found no evidence of an overall decline in tipulid populations, as measured by the abundance of their larvae in the soil. Although these data are not from standardised long-term monitoring schemes, if they reflect true population changes they would align earthworms with recent studies on other invertebrate declines which have been recorded globally [2–5]. Based on analyses of occurrence data, there is no evidence

for strong trends in an index of occupancy of craneflies in the UK from 1970–2015 [14], in line with our conclusions for changes in tipulid abundance.

Ranging between 1.6% to 2.1% per annum, the magnitude of the earthworm decline was similar between analyses, whether weighting the data to account for variation in sample extent between studies or not and using different approaches to correct for methods between studies. This is equivalent to a 33% to 41% decline over 25 years which compares with aerial insect biomass declines in Germany of 76% over 27 years [1], and of aerial and surface arthropod biomass declines of 67% in grasslands (78% abundance) and 41% in woodlands from 2008 to 2017 [41]. Annual declines of 3.8% for macromoths, 5.0% for beetles and 9.2% for caddisflies have been reported from 1997 to 2017 in the Netherlands [42]. In the UK, the population index for butterflies in the wider countryside has declined by 22% from 1976 to 2020 [43], whilst aphid populations were largely stable, and moth populations declined by 31% in Great Britain from 1969 to 2016 [12]. The latter included declines of 51% in broadleaf woodland, 43% in the uplands, 44% in urban habitats and 28% in improved grassland from 1968–2016 [44]. Declines in aerial insect abundance have been reported at some English sites, but not others [45, 46]. Overall, it appears that the putative magnitude of earthworm declines suggested by our analysis are similar to that of other declining insect groups in the UK, if less than declines reported from continental Europe. These patterns contrast with aquatic invertebrate populations where declines between the 1970s and 1990s particularly of caddisflies, mayflies and stoneflies have been reversed more recently [14], probably due to improved water quality between 1991 and 2011 [47], and mixed trends in the occurrence of many terrestrial insect groups [14]).

Any decline in earthworm populations could be contributing to overall declines in ecosystem function and biodiversity [20] as they are vital for a range of ecosystem services [48] and are keystone prey for many species [16]. If robust, our results identify a previously undetected biodiversity decline that would be a significant conservation and economic issue in the UK [49], and if replicated elsewhere, internationally, but we are not aware of any long-term earthworm population monitoring trends to compare our results with. Maggi & Tang [50] estimated that the effects of residue from pesticide use in farmland has caused a global decline in earthworm population size of less than 5%, but suggested certain geographic and agricultural areas, such as in South America, and East and Southeast Asia, have had more than a 50% decline. Stroud [23] suggested that 42% of farm fields have an absence or rarity of epigeic and/or anecic earthworms, but presented no long-term data.

Earthworm populations and trends differed between habitats, although given the variable trends between habitats, it is more difficult to generalise about spatial variation in abundance as this may be time-dependent. The unweighted model did not identify differences in earthworm abundance between habitats, but the weighted model suggested that earthworm abundances were lowest in unenclosed and wooded habitats (Table 3). Breaking these down into finer-resolution habitats suggested that the highest earthworm densities were from urban greenspace and pasture, which could be linked to fertilisation, with the lowest densities from industrial and coniferous/mixed woodland, potentially due to disturbance and acidic soils (see below). Densities from moorland varied widely between the weighted and unweighted models, being low in the weighted model (Table 4). This variation perhaps reflects the relatively small number of studies and their widely variable scale from this habitat. Previous studies have suggested that farmed habitats, particularly pasture, tend to have higher earthworm densities than woodland [36, 51, 52], and also that unenclosed habitats had the lowest earthworm abundance, most likely due to the acidic soils associated with this habitat [32, 51, 53], potentially suggesting that the weighted model for earthworms may be the most ecologically representative.

There was significant variation in trends between habitats. The greatest declines in earthworms were in woodland, followed by farmland, whereas unenclosed habitats showed a non-significant increase in earthworm abundance over time (Table 5). On closer inspection of the finer habitat categories, the declines were most apparent in broadleaved woodlands rather than coniferous woodlands (where densities were in any case low), whilst on farmland, declines tended to be strongest on pasture and grass habitats (where trends differed significantly from zero in the unweighted model; Fig 7) compared to arable habitats where negative trends did not differ from zero (Table 6; P = 0.42). Human habitats also showed evidence of significant declines in earthworm abundance over time, particularly when split into the finer categories of industrial and greenspaces. The industrial habitats included proximity from factories and ex-coal mines investigating the effects of pollutants and industrialisation on earthworm abundance [54, 55]. There was no evidence for significant variation in tipulid abundance and trend between farmland and unenclosed habitats, although our ability to investigate this was probably hampered by a lack of data.

## Mechanisms of change

Our identification of potential declines in earthworm populations in broadleaved woodlands is novel, with Model 4 suggestive of large declines of 5.7% per annum, equivalent to a 77% decline in abundance over 25 years. This suggests that earthworms could be added to other biodiversity groups in the UK shown to be declining in woodland, including moths [44, 56], butterflies and birds [43] along with arthropods in Germany [41]. These declines in woodland biodiversity appear greatest in south-east England and contrast with more positive trends in the north [11, 29], potentially consistent with a climate change mechanism. Given the sensitivity of earthworm populations to soil moisture [22, 57], it is possible that the drying out of woodlands as a result of hotter, drier summers in the south, potentially exacerbated by large-scale drainage of the countryside, could be contributing to this pattern. This would certainly fit with patterns of declining earthworm feeding birds such as the song thrush (*Turdus philomelos*) and mistle thrush (*Turdus viscivorus*) in southern England, whose abundance is linked to earthworm populations and where declines have been linked to increased drainage [27, 58]. At the same time, however, a wide range of other pressures are impacting woodlands, including increases in the abundance of non-native deer affecting woodland structure [59], diffuse pollution from agricultural run-off known to reduce earthworm populations [60], and changes in woodland management. Noting that the mechanisms underpinning moth declines in woodland remain uncertain [44], we suggest that more specific sampling of earthworm populations across a gradient of woodlands is required to test the extent to which large-scale patterns match that expected and can be related to different environmental factors.

Our analysis also provides evidence of strong declines in earthworm populations on farmland, particularly on pasture and grass categories, whilst densities in arable farmland were lower than pasture. Again, these patterns are consistent with the general pattern of biodiversity declines on farmland [33]. Agricultural intensification has probably affected earthworm populations through a range of mechanisms. There is widespread evidence that pesticides increase mortality, reduce fecundity, and decrease overall earthworm community biomass and density [61] with negative effects on earthworms apparent across 85% of parameters tested [62]. Given the widespread nature of these lethal and sublethal effects, and the widespread presence of pesticide residues across soils [63], these chemicals could also be affecting earthworm populations in other habitats, potentially contributing to declines in urban habitats and woodlands, particularly if fragmented within an agricultural matrix.

Direct cultivation by ploughing can kill earthworms, expose them to predators and change their habitat [64, 65]. Although the evidence is mixed, depending on soil-type, climatic conditions, and tillage practice, declines in abundance can range from being 2 to 9-fold [66]. Effects also vary between species, with deep-burrowing anecic species most likely to decline, whilst endogeic species may increase through increased nutrient availability [66]. This would account for the relative lack of epigeic and/or anecic earthworms on farmland in the UK [23]. Decreased soil disturbance, e.g. through non-inversion tillage or direct drilling, can increase earthworm abundance on farmland but can take up to three years from changing treatment [64, 67]. A strong impact of cultivation upon earthworm populations would certainly account for the contrast in earthworm densities between arable and pasture/grass habitats. This means that converting pasture to arable use would be expected to cause a reduction in earthworm abundance [67, 68], with likely wider impacts on farmland biodiversity [69].

Another key component of farming is fertilisation, which can have a positive impact on earthworm populations if using organic fertiliser [70, 71]. There is good evidence that the presence of dung or manure increases earthworm abundance [72–74], but, long-term fertilisation, particularly by intensive use of inorganic fertilisers, can be detrimental [70, 75]. Changes in agricultural practices from the use of natural, organic manure to concentrated inorganic chemical fertilisers over time may therefore have contributed to reductions in earthworm populations across habitats, particularly given the potential for diffuse pollution. Soil compaction from livestock may also reduce earthworm abundance, particularly of surface dwellers [76].

Our results suggested declines in earthworm populations in human-dominated habitats including industrial and green-spaces. Urbanisation poses threats to soil invertebrates through reduction in soil organic matter [77]; use of household as well as agricultural pesticides [78, 79]; leaching of chemicals from factories [55]; and, in some extreme cases, dumping of sewage particularly when discharged into water courses. In industrial landscapes, heavy-metal and other pollution may be a major constraint, whilst replenishing biodiversity loss from reclaimed industrial areas, such as spoil heaps or ex-coal mines often requires sympathetic management for recolonising invertebrates [77, 80]. Species such as earthworms can take a significant amount of time to recolonise such degraded habitats due to limited food supply [54, 70]. Despite this, greenspaces within urban areas, such as playing fields, may be refuges for soil invertebrates [22], contributing to their benefit for other taxa, such as birds [81].

Tipulids are agricultural pests long subject to control [82], and it is thought that organic practices have contributed to a resurgence in tipulid abundance [83]. They accelerate soil nitrogen fixation, can have impacts of greenhouse gas emissions [84], and are a vital food resource for birds [16]. Although, tipulid populations may suffer from hotter, drier climates particularly in the south east of the UK [83], and at the southern margins of upland peatlands [85], and can be sensitive to extreme weather events including flooding [82, 86], we found no evidence for long-term declines in abundance to date, matching occurrence trends since the 1970s [14]. Pesticides have been shown to successfully control tipulid numbers and a switch to winter cereals has lessened the impact on crop damage, however, grass as rotation cropping can increase the numbers which increases the susceptibility of subsequent spring barley [83]. Given these potentially mixed effects of different drivers, our uncertainty in their long-term trend is unsurprising, and suggests that, as with earthworms, further long-term monitoring of their populations would be desirable.

## Understanding the caveats and biases of the results

Although we have attempted to describe potential changes in earthworm and tipulid populations through time, the data that we have used to do this are not from a structured monitoring

scheme. Instead, they are unstructured data from various studies and locations, set-up with differing research questions in mind, and with variable methods over time. We have attempted to control for some of this variation by accounting for method in the analysis, for example testing for differences in the efficacy of chemical/surface extractions versus core sampling (e.g. [87, 88]), and depth of sampling, in our models. The fact that our results appear robust to the resolution at which we have done this (comparing models 1–4 with 5–8) is encouraging.

Another major source of variation between studies is sample extent. Some studies present data from one or a small number of sites (e.g. [89]) whilst others present data from tens or hundreds of locations across a large area (e.g. [22]). The former, whilst spatially more accurate, are likely to produce more uncertain estimates due to likely greater stochasticity. Study extent appears to be decreasing over time which could skew our results (Appendix 1 in S1 File), so we present the results from both unweighted and weighted models, the latter where greater weight is given to papers which present data from a greater number of samples or survey area. The consistency or not between these two models provides a guide about the confidence we may have in the results. This consistency was good for earthworms, suggesting that the reported trends are likely to be robust, whilst there was more contrast between the weighted and unweighted models for tipulids, where the overall volume of data was much less, suggesting the trend for this group is more uncertain.

Thirdly, the locations of studies will vary through time, and there are very few repeat data included (only one study had repeat samples through time that spanned more than a decade, whilst at the 10-km square level, data were recorded from multiple years spanning over 10-years from only 10 squares). This means that any bias in the apparent trends could result from non-random variation in the types of study and data published that we have collated. For example, the accuracy of any habitat-specific trend produced is dependent on the assumption that studies are equally representative of that habitat-type through time. As noted, we have attempted to account for variation in method and sample area/extent in the analysis, but there may also be variation in the underlying motivation of particular studies through time, which could affect the results. To test for this as best we could, we separated for earthworms only, whether studies were set-up specifically to examine variation in earthworm abundance, or whether data on earthworm abundances were collected as explanatory variables for other analyses, such as habitat use of a potential earthworm predator, as we thought it possible that these two motivations could result in different study selection and therefore bias the results. However, replicating Model 3 with this included as an additional factor (Appendix 2 in S1 File), did not alter the results.

Whilst we cannot treat our results as equivalent to trends from a structured, designed monitoring scheme, as far as possible we have attempted to follow best practise with analyses of these data [40]. The additional analyses presented do not indicate, at least for earthworms, that the putative negative earthworm trends reported, are subject to biases due to sample method, sample area/extent and study design, as best as we could measure them. Unfortunately, due to lack of data collated on tipulids, we were not able to perform the same tests for this group, with long-term trends being more uncertain given the differences between weighted and unweighted models, although there was no evidence for long-term population decline in this group.

## Implications

To summarise, our results suggest that earthworm populations have declined in the UK, particularly since the 1960s, with declines most apparent from farmland (particularly pasture and grassland), broadleaved woodland and human habitats (particularly greenspace and industrial

categories). Lower earthworm densities in pasture compared to arable habitats suggest the conversion of pasture to arable use would also reduce earthworm abundance. Tipulid trends were much less certain, but do not appear to be declining. Given the lack of soil invertebrate monitoring data, these results are highly novel, but consistent with expectations from the published literature, particularly with respect to reported large-scale biodiversity trends on farmland [33]. If possible, contemporary sampling of the studies from which data were collated could be a good way of validating these suggested trends.

A decline in earthworms would have major implications for soil health [17] and processes [18] as they are critically important in the delivery of a wide range of ecosystem functions including nutrient cycling and soil formation. Ecologically, as detritivores, they underpin many food webs as prey for birds (e.g.) [16], and mammals including badgers [90], shrews [91], moles [92] and foxes (*Vulpes vulpes*; [93]), therefore any long-term decline could have implications for these species. Certainly, the habitat use of thrushes (*Turdidae*) is associated with higher abundance of earthworms [58]. Declines in song thrush populations in the UK have been linked to earthworm abundance [57], whilst shifts in the distribution of wintering lapwing (*Vanellus vanellus*) and golden plovers (*Pluvialis apricaria*) away from lowland farmland to the coast [94], may be caused by a lack of soil invertebrate food on terrestrial farmland. Population trends of mammal predators appear mixed with potential declines in shrews and foxes but increases in badgers [95, 96], but are likely to be driven by a wide-range of factors including direct control.

Our results suggest that earthworms may be added to the list of taxa potentially negatively affected by agricultural intensification through a range of mechanisms, whilst the apparent decline in earthworm abundance from woodland adds weight to the potential for an increasing signal of biodiversity declines in UK woodlands [26], although with a more uncertain cause [44]. Conversely, there is no evidence for long-term declines in tipulid populations, despite the potential vulnerability of such populations to climate change [85, 97], and their control as agricultural pests [82]. If these patterns are robust, then further research is required to understand the drivers of earthworm declines across the UK, and to inform potential management responses. For example, positive management in farmland to mitigate the declines in earthworms could include reducing tillage by changing to non-inversion tillage or direct drilling, reducing pesticide use through organic farming, changing from inorganic NPK fertilisers to organic manure based fertilisers and increasing liming to reduce the acidity caused by fertilisation [75].

The current study highlights the need for a long-term and large-scale earthworm monitoring scheme which could be carried out by citizen/community scientists [22, 23]. Unlike previous projects, this should cover a range of habitats, particularly including farmland and woodland. Core sampling may be the most effective approach [88], for example as used by school children [22, 58], and could also provide data on other soil invertebrates [22], particularly if undertaken at multiple times of the year. If one season must be selected, then late spring and summer should be avoided when earthworm availability is reduced. Secondly, in order to understand the ecological causes and consequences of changes in soil invertebrate populations, spatial variation in their abundance could rapidly be linked to other environmental and biodiversity data, for example to test whether they are linked to spatial variations in bird populations [29]. In the short-term, one priority could be to repeat some of the surveys reported on in this paper, to provide more robust estimates of long-term change.

## Supporting information

**S1 Table. PRISMA checklist.**
(DOCX)

**S1 Fig. PRISMA flow diagram.**
(DOCX)

**S1 File.**
(DOCX)

## Acknowledgments

We would like to acknowledge the help of UEA Master's students Rhianna Wren, Evan Burdett, Abigail Hunns and Ashis Datta who assisted with the review and identification of relevant papers.

## Author Contributions

**Conceptualization:** James W. Pearce-Higgins.

**Data curation:** Ailidh E. Barnes.

**Formal analysis:** Ailidh E. Barnes, Robert A. Robinson.

**Investigation:** Ailidh E. Barnes.

**Methodology:** Ailidh E. Barnes, Robert A. Robinson, James W. Pearce-Higgins.

**Project administration:** James W. Pearce-Higgins.

**Supervision:** James W. Pearce-Higgins.

**Validation:** Robert A. Robinson, James W. Pearce-Higgins.

**Visualization:** Ailidh E. Barnes.

**Writing – original draft:** Ailidh E. Barnes.

**Writing – review & editing:** Robert A. Robinson, James W. Pearce-Higgins.

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
