## [Decision Letter · Decision Letter 0]

31 Aug 2022

PONE-D-22-22406Collation of a century of soil invertebrate abundance data suggests long-term declines in earthworms but not tipulids.PLOS ONE

Dear Dr. Barnes,

Thank you for submitting your manuscript to PLOS ONE. After careful consideration, we feel that it has merit but does not fully meet PLOS ONE’s publication criteria as it currently stands. Therefore, we invite you to submit a revised version of the manuscript that addresses the points raised during the review process.

We look forward to receiving your revised manuscript.

Kind regards,

Tunira Bhadauria, Ph.D.

Academic Editor

PLOS ONE

Journal Requirements:

4. We note that [Figures # 1a and 1b] in your submission contain [map/satellite] images which may be copyrighted. All PLOS content is published under the Creative Commons Attribution License (CC BY 4.0), which means that the manuscript, images, and Supporting Information files will be freely available online, and any third party is permitted to access, download, copy, distribute, and use these materials in any way, even commercially, with proper attribution. For these reasons, we cannot publish previously copyrighted maps or satellite images created using proprietary data, such as Google software (Google Maps, Street View, and Earth). For more information, see our copyright guidelines: http://journals.plos.org/plosone/s/licenses-and-copyright.

   1. You may seek permission from the original copyright holder of Figures # 1a and 1b to publish the content specifically under the CC BY 4.0 license.  

Reviewers' comments:

Reviewer's Responses to Questions

**Comments to the Author**

1. Is the manuscript technically sound, and do the data support the conclusions?

Reviewer #1: Yes

Reviewer #2: Yes

2. Has the statistical analysis been performed appropriately and rigorously? 

Reviewer #1: Yes

Reviewer #2: No

3. Have the authors made all data underlying the findings in their manuscript fully available?

Reviewer #1: Yes

Reviewer #2: No

4. Is the manuscript presented in an intelligible fashion and written in standard English?

Reviewer #1: Yes

Reviewer #2: Yes

5. Review Comments to the Author

Reviewer #1: The case for doing the study of long-term abundance of earthworms and tipulids in the UK, as well as hypotheses, were very well stated. The study was well designed, and the authors took reasonable care to examine the appropriate papers and assemble a data set. The models developed and statistical methods used are well matched to the data and hypotheses to be tested. The manuscript is very well written and easy to understand.

Throughout the paper, every time I had a question, it was answered a sentence or two later, e.g. lines 411-413 about declining earthworm abundance in woodlands, and then on lines 417-420, the mechanism that I thought most likely responsible--declining soil moisture due to warmer summers, was discussed.

Potential problems due to the fact that the data do not result from standardized repeated sampling at the same locations over time, are very clearly explained, so that readers can make their own judgements about the validity of the study. The authors have taken all needed cautions and and used all appropriate adjustments to statistical methods to account for these potential shortcomings, and I believe the overall result is of significant value and a good advance in understanding of long-term changes in ecology, especially with respect to earthworms.

Suggestions for further research and ecological implications of declining earthworm populations, and management actions that may help to mitigate the problem are nicely stated at the end of the paper.

Reviewer #2: Review of “Collation of a century of soil invertebrate abundance data suggests long-term declines in earthworms but not tipulids”

Here the authors compile an impressive dataset of earthworm data in the UK and examine trends through time. As soil inverts are often overlooked, I liked the ambition of this paper and think it will have many readers and users of the dataset. I have lots of small comments rather than few big ones. The methods are not sufficiently explained at the moment, and the results are not always clearly presented. The stats need another look but I don’t expect the patterns to change. Overall it needs a bit of tidying, which I am confident the authors can do.

Comments (chronological rather than reflecting importance)

Line 51: “In other areas” – vague, I’d reword.

Line 65-67 - Awkward sentence. Suggest rewrite.

Line 67 – “Building on the approach of Robinson & Sutherland’ – I am not sure this statement is really fair to all the other authors of studies compiling datasets from data within past studies. It’s a very popular approach right now (e.g., van Klink et al., all the BioTime stuff, PREDICTS stuff etc.., without even getting into the core meta-analysis literature), so seems a bit unfair to only highlight one study of one the co-authors of the present study.

Line 71 – not sure what “one” refers to – a country?

Line 79- the inclusion of tipulids came as a bit of a surprise to me, as I wouldn’t regard them as obvious soil inverts. Maybe you want to say something about why you chose earthworms and tipulids for your study.

Line 80 – the hypothesis that trends vary between habitats is rather weak, without specifying how they vary. If you didn’t have any stronger prediction, then I would just drop this. Or just be frank and say you explored the differences between habitats.

Line 86 – Make clearer whether you searched for papers within a time frame

Line 106 - I would more clearly state your inclusion criteria – you say you “check if they contained relevant invertebrate data” but what would that look like. Data on abundance or biomass or xxx on earthworm (one or more species??) or tipulidae, in one or more years at one or more sampling sites, collected using standardized sampling methods across sites/years?? Its rather vague at the moment and so your search would not be repeatable.

Line 123 – I am not sure what you mean by “ correlations of these against ….”. Was this using the other studies that did present both total and adult numbers? Are you sure you used correlation? I can’t see how that would work – I could guess if you said regression. (Later, I see eqn 1 and 2, so it is regression). Please expand and clarify anyhow.

Line 129 Density per m2 of soil sampled, I assume (but would that be a volume in m3?)? Or m2 land surface area of sampled? But then that wouldn’t account for sampling depth. Again, you just need to clarify a bit more.

Line 134 – For this equation to be useful for others, we need to know the units of the wet biomass

Line 131-134 – Based on this equation, I am guessing you did regression, and also guessing you assumed a normal distribution. For a response like ‘total number’, it would be more common to use a Poisson distribution to bound the predictions above zero. Did you consider this? Can you justify your current approach? Also was this a standard regression or did it include random effects? I can imagine that each study reported on multiple sample sites, so a study-level random effect (intercept and slope) would be warranted.

Line 157 - I think you mean that Year was centred so that 1960 was zero, so that the intercept of the fitted relationship represented the predicted abundance in 1960.

Line 158 – so were these data points just excluded from the analysis or included as an ‘other’ groups?

Line 160 – I am not quite sure what your response in your model was. Number or Number per m2?

I am thinking, based on line 139, it was the latter. But then I don’t understand how you modelled a decimal with negative binomial.

I feel it would have been neater to have the number of individuals as the response, and then include sampling extent as an offset term. This would mean you keep a nice count variable as your response, but model the variable in sampling effort as an offset term (basically moving the denominator of the response to the right hand side of the equation). This is also much neater than doing any weighting since results from weighted model can be driven by a few studies given a large weight.

See this paper for an example

Dealing with Varying Detection Probability, Unequal Sample Sizes and Clumped Distributions in Count Data | PLOS ONE

https://doi.org/10.1371/journal.pone.0040923

Table 2 – where did the habitat info come from? Did you overlay the study coordinates onto a land cover map? Or did you try to infer based on the study description of its habitat.

Line 188 – What was the median time span and number of sampling years across studies? Also the mean and IQR of sampling sites per study?

Line 213 – For the coefficients of factors of multiple levels, I am not sure what is being compared with what. E.g., line 213 Fewer earthworms were collected in summer compared with when? I am guessing Autumn since that is the season missing in Table 3. If so, any reason why you compared each season to Autumn in particular (apart from it being the R default!)? Similar comment applies for the other factors here like habitat and method. You don’t mention any post-hoc multiple comparisons – which is fine. But you still need to justify your reference levels for comparison for each variable e.g., the reference level could be the most common level or it could be one where you hypothesize high or low values.

Figure 2 is missing some regression lines

Table 5 – bit unclear what numbers are being presented in this table. Are these the interaction term coefficient? Or the trends within each subset? but this wouldn’t naturally fall out the model unless you removed the intercept. Again, if the former, make the reference levels clear.

Figure axes labels of “Estimate” aren’t very informative. Be more specific.

It would be nice to see Fig 3 with study-level random slopes for year to get a better idea of whether the overall mean is really a consistent pattern at the study level too. This plot could go in the SI. The overall mean decline isn’t massively convincing looking at the data points here! If anything, it looks non-linear – but that might be caused by between-study differences, which is why I suggest the random slopes.

I think Outhwaite’s paper (ref 14) also included tipulidae – how do your results compare with hers? Worth a mention.

Literature

Key papers are not cited, including some very high-profile ones on earthworm data and distribution. It would only be fair to these authors who already compiled a large database of earthworm data to cite them:

Phillips, H.R.P., Bach, E.M., Bartz, M.L.C. et al. Global data on earthworm abundance, biomass, diversity and corresponding environmental properties. Sci Data 8, 136 (2021). https://doi.org/10.1038/s41597-021-00912-z

Phillips et al. Global distribution of earthworm diversity, Science, 371, 6525, (2021). /doi/10.1126/science.abe4744

6. PLOS authors have the option to publish the peer review history of their article (what does this mean?). If published, this will include your full peer review and any attached files.

Reviewer #1: No

Reviewer #2: No

---

## [Author Response · Author response to Decision Letter 0]

24 Jan 2023

Please see attached document Response to Reviewers.

---

## [Editor Report · Decision Letter 1]

8 Feb 2023

Collation of a century of soil invertebrate abundance data suggests long-term declines in earthworms but not tipulids.

PONE-D-22-22406R1

Dear Dr. Barnes

We’re pleased to inform you that your manuscript has been judged scientifically suitable for publication and will be formally accepted for publication once it meets all outstanding technical requirements.

Kind regards,

Tunira Bhadauria, Ph.D.

Academic Editor

PLOS ONE

Additional Editor Comments (optional):

Reviewers' comments:

<quillbot-extension-portal></quillbot-extension-portal>

---

## [Editor Report · Acceptance letter]

17 Feb 2023

PONE-D-22-22406R1 

Collation of a century of soil invertebrate abundance data suggests long-term declines in earthworms but not tipulids. 

Dear Dr. Barnes:

I'm pleased to inform you that your manuscript has been deemed suitable for publication in PLOS ONE. Congratulations! Your manuscript is now with our production department. 

Kind regards, 

on behalf of

Dr. Tunira Bhadauria 

Academic Editor

PLOS ONE